

# Influence of heat index on regional mortality in Europe

Daniel Lee[1] and Thomas Brenner[1]

[1]Philipps-University Marburg, Deutschhausstr. 10, 35037 Marburg, Germany

*Correspondence to:* Daniel Lee (leed@staff.uni-marburg.de)

**Abstract.** Numerous studies of single locations have shown that high perceived temperatures have negative health conse-
quences – including higher mortality – for humans. Others have confirmed that the frequency of such dangerous heat events
will increase in the future. This study extends these insights by examining the relationship between heat index, a measure of
the physiological effects of heat, and mortality for a large area (Europe) on a regional scale. The results show that, in Europe
as a whole, the heat index has a significant influence on mortality. Furthermore, this influence is heterogeneously distributed
throughout different regions in Europe. In light of Europe's changing climate, this knowledge can serve as a basis for policies
to mitigate the effects of heat waves in the future.

## 1  Introduction

The global climate is changing, resulting in manifold consequences for natural and human systems. These changes result in a
net increase in global mean temperature, but the incrase will be unevenly distributed in time and space (e.g., Vose et al., 2005;
Diffenbaugh and Ashfaq, 2010; Diffenbaugh and Scherer, 2011; Alexander and Arblaster, 2009; Meehl et al., 2009; Smith
et al., 2005; Sherwood et al., 2008). The changes will make adaptation in ecological and economic systems necessary, as well
as in the behavior of individuals in order to adapt to the new circumstances (Intergovernmental Panel on Climate Change,
2014).

Being subject to higher heat loads creates problems for several human systems. It affects them directly by modifying the
water household and reducing agricultural yields (Calzadilla et al., 2010). It also reduces economic productivity by forcing
workers to increase break frequency and work more slowly (Kjellstrom et al., 2009). More directly, additional heat load can
cause health problems, especially for the sick and the elderly. In 2003, for example, a heat wave was responsible for the
deaths of more than 10,000 people in France alone (Poumadère et al., 2005). The frequency and intensity of extreme heat
events is projected to increase in the future (Beniston, 2004; Intergovernmental Panel on Climate Change, 2014). These facts
demonstrate the relevance of research into the effects of heat events, as well as into strategies for mitigating their negative
effects.

The increasing occurrence of hot weather in Europe in the context of climate change has been amply demonstrated (e.g.
Lee and Brenner, 2015). A number of previous studies have shown the impact of high temperatures for specific locations (e.g.
Burkart et al., 2011; Basara et al., 2010). However, a thorough analysis of the regional effects of heat on mortality for a large
area has yet to be done. This study adds to the existing literature by evaluating the regional effects of hot weather on mortality



in Europe. It is proposed that the effect of hot weather on mortality can be identified on a regional level and, furthermore, that the effect of hot weather on mortality varies regionally. In light of the increasing frequency and intensity of heat events in Europe, these insights into the dynamics between heat and mortality can help quantify the risk in various regions associated with high temperatures, identify regions with successful strategies for mitigating these effects and aid in the adaptation and

further development of such strategies.

We use the heat index (HI), a common metric in the meteorological and health communities for physiologically relevant heat load on humans, to investigate the effects of hot weather on regional mortality. A generalized method of moments (GMM) panel regression is applied, with regions partitioned according to baseline climatological, demographic and economic criteria. By examining different types of hot weather events, we find that HI has a consistent, significant positive effect on mortality.

The mechanisms through which HI affects mortality, however, appear to vary longitudinally across regional subgroups.

## 2   Theoretical background

### 2.1   The influence of heat on human health

Heat affects the human body through a variety of mechanisms. For example, thermal stress can lead to cardiovascular mortality (Burkart et al., 2011). In recent decades, heat waves have been associated with pronounced spikes in mortality, especially among

the elderly (Poumadère et al., 2005). Urban areas are affected disproportionately by heat waves due to the urban heat island (Basara et al., 2010). These facts, combined with the global processes of climate change, population aging, and urbanization, show that heat plays an increasingly important role in human health.

Yet physiologically, temperature alone is an insufficient metric for heat stress in humans. In fact, the quantification of heat load on humans is a complex undertaking, with the result that there is no standard, unambiguous descriptive metric that can be

applied across the board to describe the effect of heat on the human body (Perkins and Alexander, 2013). Heat load is caused by the combination of latent and sensible heat. The body rids itself of thermal energy partially through the evaporation of sweat. If humidity is high, this process occurs less efficiently, thus increasing the physiological heat load. For this reason, humidity is often incorporated into metrics that measure heat load on the human body.

A common metric for human heat load that is used in controlled environments is the wet bulb globe temperature (WBGT),

because it incorporates temperature, humidity, wind speed and radiation, all of which are relevant for the diffusion of heat from the body (International Organization for Standardization, 2010; National Institute for Occupational Safety and Health, 1986).

While WBGT provides highly accurate assessments of the heat load on the human body, it is impossible to measure on a regional or global scale. This is due to the fact that the many variables it requires fluctuate widely even over short distances. In the field of meteorology, several metrics are available to quantify heat load, issue warnings, etc. (Fröhlich and Matzarakis,

2015). A widely used metric is HI (Anderson et al., 2013). This metric enjoys wide use in the health and meteorological communities because it can be computed from the variables humidity and temperature (e.g. Perry et al., 2011; Kysely and Kim, 2009; El Morjani et al., 2007; Burkart et al., 2011; Basara et al., 2010). Not only are these the most important variables



when measuring physiological heat load, but they are also much more spatially homogeneous than the other variables required to compute WBGT and are readily availabe in the outputs of all numerical weather prediction models.

It can be expected that Europe will be strongly affected by increased HI in the future. Not only has HI risen over the past decades, but also the likelihood and intensity of extreme heat events has increased significantly in Europe (Lee and Brenner, 2015). Average HI for all of Europe is higher this century than in the last (see figure 1).

In contrast to the uniform increase in HI across Europe, the demographic and economic situations that contribute to the effect of HI on humans vary widely. It is to be anticipated that some regions are better adapted to mitigating the effects of high HI than others. In light of these facts, we hypothesize:

1. When viewing Europe as a whole, high HI and extreme HI events significantly increase mortality.

2. Regions in Europe with more elderly populations will be more strongly affected by high HI than regions with younger populations, due to the increased vulnerability of elderly people to cardiovascular mortality when subjected to heat load.

3. Regions in Europe with higher GDP per person will be less strongly affected by high HI than regions with low GDP per person, because people in regions with high GDP per person will generally have more options to mitigate the effects of hot weather. For example, a higher proportion of workers work in modern buildings in these regions and private housing will probably have higher standards.

4. Regions in Europe with high average HI will be affected less strongly by high HI events than regions with low average HI, because their populations use strategies for dealing with heat load (architectural styles which collect cool air in buildings, air conditioning, clothing styles, etc.).

# 3 Material and methods

## 3.1 Data

### 3.1.1 Weather data

HI values were obtained from a global dataset derived from the ERA-Interim weather reanalysis (Berrisford et al., 2009) available in high spatiotemporal resolution for the entire planet (Lee, 2014). It is provided on a 75°x75° latitude-longitude grid and aggregated to daily minima, means and maxima for each grid point on the globe for the years 1979–2013.

These data were classified according to crossed danger thresholds using GRASS GIS (GRASS Development Team, 2015). We used the thresholds provided by United States National Weather Service (National Weather Service, 2014). They were chosen because of their direct association with HI and their focus on its effects on human health. This produced the four levels of dangerous heat exposure found in table 1.

The HI danger categories were regionalized into each NUTS 3 geographic region in Europe using geographic boundaries obtained from the Geographical Information System at the COmmission (GISCO) (EuroGeographics, 2010). They were ag-





**Figure 1.** Change in average HI over Europe from the 20th to the 21st century.

gregated to a single value per region and time step in the relevant period by computing the spatially weighted maximum HI for each day using GRASS GIS (GRASS Development Team, 2015).



**Table 1.** Heat index danger thresholds, their associated prefixes in the paper's nomenclature and the danger levels they signify.

| Danger level | Suffix | Threshold |
| --- | --- | --- |
| Caution | caution | 80.0 |
| Extreme caution | ext_caution | 91.0 |
| Danger | danger | 103.5 |
| Extreme danger | ext_danger | 126.0 |

**Table 2.** Variable prefixes used in the paper and the metrics they refer to.

| Prefix | Criteria |
| --- | --- |
| nrun | Number of waves |
| longrun | Length of longest waves |
| cross | Times threshold crossed |

As is the case with heat in general, the effects of HI are multifacetted and not readily classified. The absolute HI, as well as the number of consecutive hot days and the number of heat waves in general, can all be of interest and are used to varying extents in different heat metrics (e.g. Meehl and Tebaldi, 2004; Fischer and Schär, 2010; Russo et al., 2014, etc.). We defined heat waves as at least five consecutive days on which the HI exceeded the chosen threshold. Using this definition, we computed

three different metrics to describe heat waves for each danger level (see table 2).

The four danger levels for each of the three yearly metrics produced 12 different measures of the annual heat exposure. These were calculated using numpy (van der Walt et al., 2011) and pandas (McKinney, 2010). To avoid cocorrelation, we used the differences between adjacent categories, rather than the absolute number of times each threshold in that category was exceeded. Thus each measure contains the number of events in its category which exceeded the number of events in the previous danger

category.

In the following text, each measure is denoted as the combination of the metric and danger threshold. For example, *nrun_caution* denotes the number of times in a year in which the maximum HI exceeded the caution threshold on at least five consecutive days, whereas *nrun_ext_caution* denotes the number of times in a year in which at least five consecutive days had a maximum HI that exceeded the extreme caution threshold, subtracted by the number of times in the same year in which at least five

consecutive days had a maximum HI which exceeded the caution threshold.

### 3.1.2 Demographic data

Demographic data was obtained from Eurostat containing annual mortality (Eurostat, 2014a), population by broad age groups (Eurostat, 2014b) and GDP (Eurostat, 2014c) for all NUTS 3 regions for the period 2000–2012. The demographic data was



**Table 3.** Analyzed subgroups and the criteria used for sorting regions into them.

| Group | Criteria |
|---|---|
| All regions | None |
| Regions with young populations | <0.18% of population is elderly |
| Regions with elderly populations | $\geq$ 0.18% of population is elderly |
| Regions with low GDP | Annual GDP per person <23300 € |
| Regions with high GDP | Annual GDP per person $\geq$ 23300 € |
| Regions with low mean heat index | Mean heat index <49.3 |
| Regions with high mean heat index | Mean heat index $\geq$ 49.3 |

mapped to the NUTS 3 geographic regions using NUTS ID and GISCO keys from Eurostat's metadata server Reference And Management Of Nomenclatures (RAMON) (Eurostat, 2013).

Mortality and GDP data were normalized in each region by that region's population. For the calculation of elderly proportion individuals over 64 years of age were considered elderly. The data was also cleaned by removing time points at which any data were missing and by removing duplicated regions. Finally, the demographic data was joined with the HI data and all following steps were conducted in the statistical analysis environment R (R Core Team, 2014).

### 3.2 Regression approach

We conducted a panel GMM regression, which allows testing for causality through the use of instrument variables (Croissant and Millo, 2008). Following this practice, we used lagged variables as instruments. Time and regional fixed effects were included in the estimations.

We analyzed the idiosyncratic effects predicted in our hypotheses by partitioning the data into different groups. Regions were partitioned according to elderliness, GDP and mean HI, as shown in table 3. Regions were partitioned at the median of the variable in question in order to create groups with similar sizes.

Results were cleaned and formatted for interpretation using Wickham (2009); Bivand and Rundel (2015); Bivand and Lewin-Koh (2015); Hijmans (2015); Pebesma and Bivand (2013); Harrell et al. (2016); South (2011).

### 4 Results and discussion

Overall, this study confirms the proposed hypotheses. High HI has a significant, positive relationship with mortality in almost all cases. In the following sections, the results are discussed in the context of the individual hypotheses. The complete results for all regressions are provided in the appendix.



**Table 4.** Panel GMM regression results for mortality as a dependent variable. Results are shown for each variable with p-values in brackets.

|  | All regions |
| --- | --- |
| n regions | 1054 |
| cross_caution | **3.635e-06 (0.006591 \*\*\*)** |
| cross_ext_caution | **4.769e-06 (0.03432 \*)** |
| cross_danger | **3.564e-06 (0.02359 \*)** |
| cross_ext_danger | **5.789e-06 (0.006443 \*\*\*)** |
| longrun_caution | 2.351e-06 (0.5193) |
| longrun_ext_caution | **1.176e-05 (0.03325 \*)** |
| longrun_danger | **1.092e-05 (0.04302 \*)** |
| longrun_ext_danger | 1.233e-05 (0.08342) |
| nrun_caution | **8.76e-05 (9.741e-06 \*\*\*)** |
| nrun_ext_caution | 4.203e-05 (0.2391) |
| nrun_danger | 2.832e-05 (0.2259) |
| nrun_ext_danger | 1.66e-05 (0.5065) |

## 4.1 Hypothesis 1: HI events lead to increased mortality in Europe

Our first hypothesis proposed that in Europe, as a whole, HI events have a positive relationship to mortality.

Table 4 provides clear confirmation of this hypothesis. All measures for regional HI are positively associated with mortality. The null hypothesis can be rejected for seven out of the twelve tested variables.

The results for the different variables show interesting variations. While the length of a heat wave and the number of heat waves have insignificant effects on mortality for some heat thresholds, the number of times in a year that the threshold is crossed significantly affect mortality in all cases. In the case of the number of heat waves the category "caution" is highly significant.

Clearly, high HI leads to greater mortality, as shown in the significance of especially the *cross* variables. However, the lower significance of especially the *longrun* variables suggests that this effect can be observed independently of the temporal coherence between high HI events.

Additionally, the high significance of *nrun_caution* suggests that heat waves are related to mortality through mechanisms above and beyond the simple crossing of danger thresholds. A large number of heat waves that exceed the "caution" threshold has a highly significant relationship with mortality, whereas the length of these waves does not. This is in agreement with health literature in suggesting that there is a behavioral risk component involved in mortality associated with high HI. These results indicate that if a heat wave lasts a substantial amount of time, vulnerable individuals can adapt, most likely due to their awareness of the danger or sensitivity to the discomfort caused by the weather. However, if heat waves occur repeatedly,



**Table 5.** Panel GMM regression results for mortality as a dependent variable. Regions are sorted into groups using criteria based on elderly proportion of population. Results are shown for each subgroup and variable combination with p-values in brackets.

|  | Regions with young populations | Regions with elderly populations |
| --- | --- | --- |
| n regions | 581 | 675 |
| cross_caution | **5.953e-06 (0.009168 \*\*\*)** | **6.879e-06 (1.782e-05 \*\*\*)** |
| cross_ext_caution | 8.708e-08 (0.9765) | 6.047e-06 (0.07236) |
| cross_danger | **4.171e-06 (0.0122 \*)** | **8.308e-06 (0.007034 \*\*\*)** |
| cross_ext_danger | 4.494e-06 (0.1189) | 8.537e-07 (0.7566) |
| nrun_caution | 2.188e-05 (0.5534) | **0.0001149 (1.101e-07 \*\*\*)** |

this awareness decreases, causing individuals to engage in less cautious behavior and thus increase their vulnerability to HI-based mortality. The low significance of heat waves with higher danger classifications could be a result of the low number of observations in these categories, or it could be due to the fact that the magnitude of the HI event causes individuals to adjust their behavior accordingly.

Therefore, we when examining Hypotheses 2 to 4 re restrict our analysis to the *cross* variables and the variable *nrun_caution*.

## 4.2   Hypothesis 2: HI events affect elderly populations more strongly than young populations

Our second hypothesis proposed that elderly populations are affected more strongly than young populations by high HI events.

    Table 5 confirm this hypothesis. While both groups are affected negatively by high HI - disregarding insignificant variables, all influences of HI on mortality are positive - three of five coefficients are large for regions with more elderly populations and
we obtain three compared to two significant impacts for those regions.

    The difference between the two subgroups of regions is strongest for the number of heat waves that cross the caution level (*nrun_caution*). Younger populations seem to be more robust towards heat waves, especially if the heat is not too high, while older populations seem to be more harmed by longer periods of heat.

    With respect to the number of occasions at which heat crosses a certain threshold there is no clear tendency. Heat events in
general seem to affect regions with more or less elderly population in a similar way. Hence, hypothesis 2 is only confirmed for the number of heat waves.

## 4.3   Hypothesis 3: HI events affect regions with low GDP per person more strongly than regions with high GDP per person

Our third hypothesis proposed that regions with high GDP per person would be more robust against high HI events than regions
with low GDP per person. This hypothesis is not confirmed by our analysis.



**Table 6.** Panel GMM regression results for mortality as a dependent variable. Regions are sorted into groups using criteria based on mean GDP per person. Results are shown for each subgroup and variable combination with p-values in brackets.

|  | Regions with low GDP | Regions with high GDP |
|---|---|---|
| n regions | 661 | 693 |
| cross_caution | **3.812e-06 (0.01733 *)** | **7.299e-06 (1.295e-05 ***)** |
| cross_ext_caution | 2.476e-06 (0.2852) | **2.459e-05 (1.59e-05 ***)** |
| cross_danger | **4.799e-06 (0.005163 ***)** | 9.986e-07 (0.7655) |
| cross_ext_danger | **5.283e-06 (0.01552 *)** | 2.475e-05 (0.06467) |
| nrun_caution | **7.571e-05 (0.0006498 ***)** | **0.0001386 (6.205e-08 ***)** |

As shown in table 6, *nrun_caution* has highly significant, positive effects on mortality, regardless of GDP per person. For the *cross* variables we obtain three significant effects for the low GDP regions and only two significant effects for the high GDP regions. However, in most cases the coefficients are higher for the regions with high GDP.

A closer look at the results for regions with high GDP provides important insights: First of all, it is notable that only measures at danger levels of "caution" and "extreme caution" are significant. HI events above this level have no significant influence on mortality, indicating that at this point, the population takes countermeasures against heat load. This most likely does not take place when the less dangerous thresholds are crossed. Nonetheless, these thresholds are associated with health risks, and these risks take their toll on the population. Thus, although the population is on a whole more robust against high HI, seemingly low-risk HI events are not taken seriously and thus lead to higher mortality. Hubris in the face of nature is not an unfamiliar characteristic of regions with high economic productivity.

Regions with low GDP per person show a different picture. They seem to be more vulnerable to extreme heat levels (crossing the danger threshold), while they seem to be better able to deal with the less extreme heat events. Maybe the measures to deal with extreme heat events are too costly to be a good option for them. The estimates in table 6 as well as in table 4 show that a higher HI leads to more deaths. Thus, economic development seem to help regions to lower the burden of the more severe high heat events. Since especially these events will become more frequent due to global climate change, we can indeed expect that regions with low GDP are more affected by this.

### 4.4 Hypothesis 4: HI events affect regions with low mean HI more strongly than regions with high mean HI

Our fourth hypothesis proposed that regions with high mean HI are less strongly affected by HI events than regions with low mean HI. Intriguingly, we find the opposite, as shown in table 7.

Regions with low mean HI exhibited highly significant relationships between only one HI variable and mortality, whereas two HI variables were highly significant in the case of regions with high mean HI. In the case of regions with high mean HI, both of these variables are associated with the "caution" danger level. The fact that none of the variables associated with higher



**Table 7.** Panel GMM regression results for mortality as a dependent variable. Regions are sorted into groups using criteria based on mean heat index. Results are shown for each subgroup and variable combination with p-values in brackets.

|  | Regions with low mean heat index | Regions with high mean heat index |
|---|---|---|
| n regions | 607 | 447 |
| cross_caution | *-1.189e-05 (0.000167 \*\*\*)* | **6.283e-06 (5.868e-05 \*\*\*)** |
| cross_ext_caution | 4.922e-05 (0.4112) | -1.572e-06 (0.4851) |
| cross_danger | -0.02235 (0.119) | -5.769e-07 (0.733) |
| cross_ext_danger | 0.0001019 (0.373) | -5.094e-07 (0.8131) |
| nrun_caution | 7.616e-05 (0.3147) | **8.905e-05 (6.638e-06 \*\*\*)** |

danger levels is significantly associated with mortality suggests that the populations of regions with high mean HI do indeed take countermeasures against heat load when extreme HI events occur. This is apparently not the case with less dangerous HI events, which nonetheless can measurably affect regional mortality. While the heat countermeasures do seem to be effective, they are likely adopted too late, at the cost of lives at the caution level.

In contrast, regions with low mean HI are fascinatingly unique among all region groups. The typically significant variables are insignificant for these regions and if they are significant (*cross_caution*) the effect has the opposite sign than usual. That means that if HI crosses the caution threshold mortality decreases. Several things can be noted here. This indicates not only the absence of common mechanisms often observed in other regions – for example, that the number of heat waves at the "caution" level is not an issue in these regions – but also that the significant variables affect the population in ways that do not occur

in other groups. It is probable that due to the low frequency of high HI events, the population is more acutely aware of their dangers and reacts especially strongly accordingly. It might also be that in regions that are usually very cold, mortality is more driven by cold than by hot periods and that a higher number of heat events in a year comes together with a lower number of extremely cold days.

## 5    Conclusions

This paper investigated the effects of heat on mortality, using a range of metrics to capture the interactions between heat events, measured using heat index, and mortality on a regional level in Europe. Using panel GMM analyses, a variety of mechanisms could be observed through which high HI influences mortality. In general, high HI events were shown to have significant, positive relationships with regional mortality, although the mechanisms at play in each group, as well as the effects they expressed, differed meaningfully.

Across all groups, the most important variables were the number of heat waves at the "caution" level (*nrun_caution*) and the number of days on which the maximum HI exceeded the "caution" level (*cross_caution*). The length of heat waves has been found to be less important.



The different effects of HI measures on mortality across groups of regions provided important insights into these interactions. Although it was shown that the frequency of high HI events is relevant due to their consistent effect of increasing mortality, the other mechanisms, such as the population's response to hot weather, are able to effectively counterbalance these influences, which was found to be the case, e.g. for extreme heat in high GDP regions. Awareness seems to be key to safely mitigating

mortality caused by hot weather.

Heat events have been observed to increase in frequency and intensity in Europe since 1979 (Lee and Brenner, 2015). It can be expected that this trend will continue in the course of climate change. As hot weather increases mortality, it is a straightforward assumption that this will be associated with higher economic costs and negative consequences for longevity. This is exacerbated by the increasing elderliness of Europe's population, as the paper demonstrates that elderly populations

are more vulnerable to heat events. Furthermore, low GDP regions seem to be especially hurt by the expected increase in dangerous heat event. Countermeasures can be taken to mitigate the effects of heat, but these cost money, either through increased expenditures or decreased productivity.

Strategies for mitigating heat are clearly important and have been shown in this and other studies to effectively decrease the health consequences of hot weather. This indicates that much can be learned from regions that deal well with hot weather. Ap-

plying these lessons in other places could save lives and improve quality of life. Investigating these mechanisms and exploring strategies for transferring them to other places is a worthwhile pursuit that should yield important results for regions in the future.

**Appendix A:  Regression results**





**Figure 2.** Influence of HI on mortality in all regions. P-values are shown on a logarithmic scale on the x-axis. Farther left shows higher significance; lines denote standard significance levels (0.1, 0.05, 0.01). The coefficient's sign is denoted by each bar's color.



**Figure 3.** Influence of HI on mortality in regions with young populations. P-values are shown on a logarithmic scale on the x-axis. Farther left shows higher significance; lines denote standard significance levels (0.1, 0.05, 0.01). The coefficient's sign is denoted by each bar's color.



**Figure 4.** Influence of HI on mortality in regions with elderly populations. P-values are shown on a logarithmic scale on the x-axis. Farther left shows higher significance; lines denote standard significance levels (0.1, 0.05, 0.01). The coefficient's sign is denoted by each bar's color.



**Figure 5.** Influence of HI on mortality in regions with low GDP per person. P-values are shown on a logarithmic scale on the x-axis. Farther left shows higher significance; lines denote standard significance levels (0.1, 0.05, 0.01). The coefficient's sign is denoted by each bar's color.




**Figure 6.** Influence of HI on mortality in regions with high GDP per person. P-values are shown on a logarithmic scale on the x-axis. Farther left shows higher significance; lines denote standard significance levels (0.1, 0.05, 0.01). The coefficient's sign is denoted by each bar's color.







**Figure 7.** Influence of HI on mortality in regions with low mean HI. P-values are shown on a logarithmic scale on the x-axis. Farther left shows higher significance; lines denote standard significance levels (0.1, 0.05, 0.01). The coefficient's sign is denoted by each bar's color.





**Figure 8.** Influence of HI on mortality in regions with high mean HI. P-values are shown on a logarithmic scale on the x-axis. Farther left shows higher significance; lines denote standard significance levels (0.1, 0.05, 0.01). The coefficient's sign is denoted by each bar's color.





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
