# Peer review of "Influence of heat index on regional mortality in Europe"

_Natural Hazards and Earth System Sciences, 2016_

## Referee Comment (RC1) · Anonymous Referee #1 · 15 Jun 2016

The manuscript presents a statistical approach for a European-wide, regional investigation on NUTS3 level how the annual heat exposure derived from Heat Index is influencing annual mortality. The topic of heat related mortality is relevant and is suitable for the scope of NHESS. I have questions in particular regarding the approach developed and tested in the manuscript, and I have further suggestions for revisions to substantially improve the manuscript.

Section 2, Hypotheses:

1. While the tested hypotheses are stated clearly, it is not exactly clear to me how you derived or developed them including the assumed underlying mechanisms. The paragraphs introducing the hypotheses (P3, line 5-8) and the four hypotheses would benefit if the concepts and literature used to derive the hypotheses are included in the

manuscript.

2. Hypothesis 4: P3, line 16: you mention adaptation in architecture and behavior as potential mechanisms to assume that HI effect on mortality is lower in high HI average countries. In the light of the room you give to physiological problems of heat and heat loads on the body in the subsection on the influence of heat on health, I wonder if the potential role of physiological acclimatization to hot weather and climate in addition to behavioral and architectural measures should also be considered in the assumed mechanisms underlying Hypothesis 4.

Section 3, Methods and Material 3. P5, line 1-15: I understand that you used 4 thresholds of HI and 3 criteria to obtain categories that account for the different frequency, intensity, and duration of heat waves, and that you used one annual value for each category as a measure for heat exposure to compare with one annual value for mortality. I have two questions:

Question 1: Your heat wave definition is "at least five consecutive days on which the HI exceeded the chosen threshold" (p5, line 4), and to avoid co-correlation you use the difference to adjacent categories in the further analysis. I am not sure if I understand this correctly. How would, for example, a persistent heat situation with a total duration of 12 days with 5 days threshold "danger", 2 days "extreme danger", and again 5 days "danger" be classified/represented in the measures of annual heat exposure? Would it be two heat waves of 5 days with danger or one heat wave? How would either affect the results of your regressions? An additional example exercise for all categories would be helpful. On p. 5, lines 11-15, you explain the combination of danger level with "nrun", but not with "cross". This would be in particular helpful as "cross" turns out to be important in the results of the regressions.

Question 2: physiological acclimatization to heat varies across Europe due to climate conditions. This is reflected in relative heat wave definitions, and, for example different thresholds are applied in heat warning systems in European countries. Did you

consider this differences in acclimatization/thresholds in your HI thresholds?

4. P 5, line 17 and following: I understand that you combine 13 years mortality data (2000-2012) with 25 years meteorological data (1979-2013). How have this different time periods of data been considered in the investigation of influence of HI on mortality? It is not clear to me whether the HI data of the years 1979 to 1999 are included in the statistical analysis at all.

5. Page 6, line 3-6: I understand that you used normalized annual mortality data (all causes) per NUTS3 region to study the relation between HI and mortality. How did you check that also other causes than heat waves could be related to additional mortality in the years of 2000-2012, and that therefore other causes than heat could have influenced the mortality? Did you use any additional criteria in your approach to define "heat-related mortality" and to attribute annual mortality to heat?

6. Same place: As you use Eurostat mortality data from 2000-2012, I wonder if regional mortality patterns of the European heat waves 2003 statistically dominate the results and conclusions on your hypotheses. Have you considered the potential effects of this particular extreme event in your analysis of HI influence on annual mortality?

7. Page 6, line 3-5: I understand that you used the GDP and related it with mortality in your HI categories. Have you checked before how the GDP is distributed over the NUTS3 regions relative to the HI? Given the European scale of analysis, I wonder if regions with lower GDP are more often located in geographical regions exposed to higher HI than regions with higher GPD. Additionally, other studies show effects of the temperature on the GDP/economic production (as an additional example: Burke et al. 2015), so that that determining cause and effects in the relation of GDP on heat mortality is a rather complex issue. I wonder if your results later in the paper regarding GDP and heat mortality rather show the effect the regional HI distribution of HI than the effect of GDP on heat mortality.

8. P6, lines 8-15: GGM regression approach: you explain in detail how you performed

the regression. Could you provide information on the goodness of fit of the regression models?

Results and Discussion:

9. Results and discussion are presented in one section, and the discussion parts are rather short and structured according to the individual four hypotheses. I miss in particular a section that critically reflects the presented approach and that discusses the results against the background of all four hypotheses. Please consider restructuring and expanding the discussion.

10. The discussion of the results on potential underlying mechanisms and interactions that might explain the results of the regressions and the varying influences of the heat exposure measures would benefit from literature that supports the argumentation and assumptions made (see examples below).

11. P7, Table 4 (and also Tables 5-7): Precise column heads are missing. What is shown? GGM Regression coefficients, p-values?

12. P7, lines 14 to p8, line 2) "This is in line with health literature suggesting that there is a behavioral risk component involved.... that if a heat wave lasts a substantial amount of time, vulnerable individuals can adapt, mostly likely due to their awareness of the danger of sensitivity to the discomfort causes by the weather. . . . this awareness decreases, causing individuals to engage in less cautious behavior and thus increase their vulnerability to HI-based mortality": please include example references from the health literature supporting these different assumptions and explanations (see general comment on discussion above).

13. P9, line 6-7: ". . . indicating that as this point, the population takes countermeasures against heat load. This most likely does not take place when the less dangerous thresholds are crossed". How can you conclude this? Additional literature, for example on behavior during heat waves, would support this assumption on underlying mechanisms.

14. P9, line 9-10: "Hubris in the face of nature is not an unfamiliar characteristic of regions with high economic productivity". This is a very strong final statement that needs a good and substantial argumentation in the lines before (see above).

15. P. 9, line 6-14: Your results show that mortality in regions with higher GDP is significantly related to the "lower" HI thresholds and in regions with lower GDP to the "higher" HI thresholds and you explain this result by resources for better adaptation to heat waves in richer regions. Again, I am wondering in how far this result reflects the European regional pattern of GDP distribution and the pattern of heat wave deaths in 2003 (see for example Robine et al., 2008).

16. P 10, line 10: "it is probable that due to the low frequency of HI events, the population is more acutely aware of their dangers and reacts especially strongly accordingly. It might also be that in regions that are usually very cold, mortality is more driven by cold than by hot periods and that a higher number of heat events…." Also this argumentation would benefit substantially from examples / references from the literature.

17. In the discussion section, I would highly welcome further critical reflections on

o How do your results agree or disagree with other studies on heat mortality from the scholarly literature that you have mentioned in the introduction and the theoretical background sections?

o What are the advantages and also limitations of your approach, what further research questions do evolve from your results (suggestions see next two bullets)?

o How could further relevant questions in heat related mortality be included in the approach? You have, for example, mentioned the urban island as a factor for heat impact (P2, line 15-16), but you did not include any proxy for urban structures or settlement in your analysis. Do you have ideas how additional proxies from EU Data could be used?

o Small-scaled variability of temperature distribution in particular in urban areas and

complex relationship of indoor/outdoor temperature on the one side and a high aggregation level of a regional approach with one observed value per Nuts3 level?

Additional references mentioned:

Burke, M.; Hsiang, S. M. & Miguel, E., 2015: Global non-linear effect of temperature on economic production Nature, Nature, 527, 235–239.

Robine, J.-M.; Cheung, S. L. K.; Roya, S. L.; van Oyen, H.; Griffiths, C.; Michel, J.-P. & Herrmann, F. R., 2008: Death toll exceeded 70,000 in Europe during the summer of 2003, C. R. Biologies, 331, 171-178.

---

## Referee Comment (RC2) · Anonymous Referee #2 · 30 Aug 2016

This paper seeks to examine the relationship between heat index and mortality in Europe. While the dataset is interesting, the authors should do a bit more work in order to show scientific significance (defined by NHESS as representing a substantial contribution to the understanding of natural hazards and their consequences). Examining each of the hypotheses:

Hypothesis 1: High HI and extreme HI events significantly increase mortality.

While the question is appropriately examined, it is unfortunately not novel. It is common knowledge that the heat index and mortality are correlated. The authors could show this as a sanity check within the methods or the appendix, but it is not novel and therefore should not be one of the main findings of the paper.

Hypothesis 2, 3: Those who are more elderly or poorer are more likely to suffer.

This hypothesis suffers from a lack of literature review in this area. It is well known how demographic variables (a much larger number than those examined in the paper) are related to vulnerability. Bradford et al. paper contains a review of literature demonstrating the relation between heat death/ illness and demographics/ other characteristics; it may be useful for the authors to review this information (as well as all of the ~100 papers cited therein) and revisit a more appropriate set of hypotheses.

Bradford, K.; Abrahams, L.; Hegglin, M.; Klima, K. A Heat Vulnerability Index and Adaptation Solutions for Pittsburgh, Pennsylvania. Environmental Science & Technology. 2015, 49(19), 11303–11311. PMID: 26333158

Hypothesis 4: High mean levels of HI reduce the population's response to extreme events.

This might make for an interesting hypothesis. I would suggest that instead of regressing simply on the mean, that one use a combination of the mean and standard deviations in order to redefine extreme events at each location. Note, Table 7 is unclear; the variable names are unintelligible, the asterisks are not defined, and the R values are missing.

---

## Author Comment (AC1) · 24 Oct 2016

The authors would like to thank both referees for their constructive and helpful comments on the article. We have attempted to address all points in the following text.

**1   Physiological acclimatization in the hypotheses**

As the referees note, the potential role of physiological acclimatization to hot weather and climate should certainly be incorporated into the assumed mechanisms underlying hypothesis 4. In this case the hypothesis can be accurately reworded as follows:

"Regions in Europe with high average HI will be affected less strongly by

[Figure]

high HI events than regions with low average HI, because their populations are better adapted for dealing with heat load (e.g. physiological acclimatization, use of architectural styles which collect cool air in buildings, air conditioning, clothing styles, etc.)."

This is supported by the literature, handled exhaustively in e.g. Cheung & McLellan (1998), and acknowledged in subsequent literature which we will gladly reference in the revised article.

Cheung, Stephen S., and Tom M. McLellan. "Heat acclimation, aerobic fitness, and hydration effects on tolerance during uncompensable heat stress." Journal of Applied Physiology 84, no. 5 (1998): 1731-1739.

**2  Category definitions**

To answer the questions about the definitions of the categories:

The persistent heat situation described in the example would be counted as one heat wave of the category "danger". Although the two days of "extreme danger" would be represented in other metrics, they would not be counted as a heat wave in and of themselves, as their total consecutive duration did not reach five days. In the reworked manuscript we are more than willing to provide example exercises to make this more clear. We agree that the variables could be described better - also with variables of the class "cross" a clearer explanation is possible and will be included in a reworked final manuscript. Here, for the sake of explanation, we can use the same persistent heat situation described in the question. 10 days would be counted to "cross_danger", while 2 days would be counted to "cross_ext_danger" for the reason that if the "ext_danger" level is reached, the heat event is counted toward that and not to the intermediate danger categories.

**3 Specific, local metrics accounting for physiological acclimatization**

Differences across Europe in acclimatization were not considered in setting HI thresholds, because these thresholds were transferred from the definitions used by the US National Weather Service. While the referee is certainly correct in recognizing the significance of local conditions in describing resilience to weather conditions, the task of describing a large geographical area with a single metric confronts one with inevitable difficulties. There are approaches that use relative climate indicators in order to warn of extreme weather events, such as the Extreme Forecast Index used by ECMWF (Laurette & Grijn, 2002). This can be considered a good strategy for dealing with climate extremes from an environmental standpoint. However, no amount of acclimatization will make climate the only relevant factor in the effects of hot weather on the human body. Lacking literature which proposes appropriate thresholds for different areas in Europe, our option was to produce such thresholds in the context of the study - which, however, would have at least the scope of a complete study in and of itself, if not more - or of leveraging a metric which could be considered robust against variable local acclimatization.

The thresholds provided by NWS are designed for use in a wide geographical area (the United States) which is comparable in area to that of Europe and are used in a wide variety of local climates. Moist, hot climates in the southeast are considered, as are hot desert areas in the southwest. More temperate areas, e.g. in the northeast, are served by the same metric with success. This naturally does not mean that the metric or the thresholds used with it are perfect, but it is tried and true in several other articles. Due to the wealth of studies cited in the paper which use it as a basis, we decided to use it without modification. The robustness and generality of the thresholds, as well as the transferability of results, overweighed the advantages of creating bespoke thresholds for the various geographical areas within the scope of the study.

Lalaurette, François, and Gerald van der Grijn. "Ensemble forecasts: can they provide

useful early warnings." ECMWF Newsletter 96 (2002): 10-18. Harvard

**4   Clarity in data usage**

The referees are absolutely correct in questioning the years of data used to analyze mortality vs. the years of meteorological data. The longer time series of meteorological data was used to classify the local climates of the areas in the study, whereas only the years in which a complete coverage of both mortality and meteorological data were available were used in computing the regressions. This will be explained more clearly in the revised article.

**5   Causes of mortality**

Although in some cases limited cause of death data was available, this was neither universally available, nor did it have the necessary thematic granularity in order to identify specific, heat-related deaths. Additionally, listed causes of death did not explicitly include heat. Therefore, many causes of death which could be linked to the weather situation - for example cardiovascular failure - would thus disappear from our study. Additionally, other causes of death which could be causally but not physiologically linked to heat would not appear in our metrics if we were to explicitly include certain causes of death. Lowered productivity, precision and concentration are symptopms of heat exhaustion, for example, but fatal industrial accidents resulting from a worker's lack of concentration would also be eliminated if deaths were filtered by attribution.

Thus we are faced with a common problem familiar from the field of econometrics - essentially, the study attempts to isolate a signal (mortality rate) propogating from a given source (weather) in a pool of noisy data. This is compounded by the heterogeneous quality of the data available. A clear attribution of the data points to a known, given cause is not possible and a relativation of the ratios on grounds of various events which took place within the study areas would introduce a degree of arbitrarity to the investigation. In light of this fact, we decided to examine the total mortality as a means of investigating heat's holistic effects on the populations of the study.

This approach of not creating any special cases within the data was also applied to the signal generated by the deadly heat wave in 2003. It is true that this was an extroadinary heat event which has been handled prominently in the literature. Removing it, however, would have resulted in the loss of an important data point and would have unnecessarily shortened the time series. Because a panel regression approach was used, removing this year would have shortened the longest time series by almost a full tenth and affected the sample size for regions with patchy data availability even more strongly. Additionally, we believe that this significant event provides insightful data for the regressions we used.

In the revised manuscript it would be possible to include a robustness check in which data from this year are eliminated in order to judge their effects.

**6   Possible effects of GDP on mortality**

The referees duly note that GDP is a known determinand of mortality, as well as weather. It is true that partitioning regions across GDP-defined boundaries separates regions with higher GDP, where lower mortality would be expected, from those with lower GDP and thus a higher expected mortality. This partitioning was in fact adopted in part in order to prevent this signal from leading to a false conclusion that HI leads to higher mortality, when in fact GDP would be the dominant determinand. By analyzing regions with similar GDP we capture the variability amongst regions with similar GDP that is caused by variations in HI.

**7   Goodness of fit of regression models**

We would be happy to update all tables to provide goodness of fit metrics on the regression models in the updated manuscript.

**8   Expansion of discussion section**

We would be glad to update the discussion section as requested by Referee #1 in order to include a section critically reflecting on the approach and synthesizing the results of all hypotheses holistically, as well as discussing possible underlying mechanisms in greater detail. We will also be glad to update the article in the next manuscript to reference literature showing the possibilities of adapting to hot weather by mitigating its effects physiologically (e.g. turning on air conditioning, drinking enough water, wearing proper clothing, etc.). These had not been included previously because of their wide acceptance as common knowledge.

Of course we are also willing to include further discussion of the approach used and in particular further questions raised by the research that can and should be pursued in future studies, as recommended.

**9   Extended literature review**

We thank the referees for their valuable references applied and would be happy to include these in the next manuscript in the context of the research we present.